# Long Term Survival in Patients Suffering from Glio-blastoma Multiforme: A Single-Center Observational Cohort Study

**DOI:** 10.3390/diagnostics9040209

**Published:** 2019-11-30

**Authors:** Daniele Armocida, Alessandro Pesce, Federico Di Giammarco, Alessandro Frati, Antonio Santoro, Maurizio Salvati

**Affiliations:** 1Neurosurgery Division, Human Neurosciences Department, Sapienza University, 00135 Roma, Italy; federico.digiammarco@gmail.com (F.D.G.); antonio.santoro@uniroma1.it (A.S.); 2IRCCS “Neuromed” Pozzilli (IS), Università Sapienza of Rome, 00135 Roma, Italy; ale.pesce83@yahoo.it (A.P.); alex.frati@gmail.com (A.F.); salvati.maurizio@libero.it (M.S.)

**Keywords:** long term survival, glioblastoma, IDH, EGFR, Ki67, p53

## Abstract

Background: Glioblastomas (GBM) are generally burdened, to date, by a dismal prognosis, although long term survivors have a relatively significant incidence. Our specific aim was to determine the exact impact of many surgery-, patient- and tumor-related variables on survival parameters. Methods: The surgical, radiological and clinical outcomes of patients have been retrospectively reviewed for the present study. All the patients have been operated on in our institution and classified according their overall survival in long term survivors (LTS) and short term survivors (STS). A thorough review of our surgical series was conducted to compare the oncologic results of the patients in regard to: (1) surgical-(2) molecular and (3) treatment-related features. Results: A total of 177 patients were included in the final cohort. Extensive statistical analysis by means of univariate, multivariate and survival analyses disclosed a survival advantage for patients presenting a younger age, a smaller lesion and a better functional status at presentation. From the histochemical point of view, Ki67 (%) was the strongest predictor of better oncologic outcomes. A stepwise analysis of variance outlines the existence of eight prognostic subgroups according to the molecular patterns of Ki67 overexpression and epidermal growth factor receptor (EGFR), p53 and isocitrate dehydrogenase (IDH) mutations. Conclusions: On the grounds of our statistical analyses we can affirm that the following factors were significant predictors of survival advantage: Karnofsky performance status (KPS), age, volume of the lesion, motor disorder at presentation and/or a Ki67 overexpression. In our experience, LTS is associated with a gross total resection (GTR) of tumor correlated with EGFR and p53 mutations with regardless of localization, and poorly correlated to dimension. We suppose that performing a standard molecular analysis (IDH, EGFR, p53 and Ki67) is not sufficient to predict the behavior of a GBM in regards to overall survival (OS), nor to provide a deeper understanding of the meaning of the different genetic alterations in the DNA of cancer cells. A fine molecular profiling is feasible to precisely stratify the prognosis of GBM patients.

## 1. Introduction

### 1.1. Background and Rationale 

Glioblastoma (GBM) is the most common primary malignant brain tumor, accounting for approximately 50% of primary brain tumors [1]. Despite the introduction of multimodal treatment protocols, the prognosis remains poor. To date, the median survival is about 16–18 months and only 3–5% of patients survive 5 years [1,2,3,4,5,6,7,8,9]. The definition “long-term survivors” (LTS) is commonly used for patients who survive more than 24 years from initial diagnosis of glioblastoma [1,4].

Glioblastomas are divided in the 2016 CNS WHO (Central Nervous System, World health Organization) into (1) GBM, isocitrate dehydrogenase (IDH)-wildtype (about 90% of cases), which corresponds most frequently with the clinically defined primary or de novo glioblastoma and predominates in patients over 55 years of age [10]; (2) glioblastoma, IDH-mutant (about 10% of cases), (referred to secondary GBM) with a history of prior lower-grade diffuse glioma and preferentially arises in younger patients; and (3) glioblastoma, NOS, a diagnosis that is reserved for those tumors for which full IDH evaluation cannot be performed [1,3,11,12].

Among these, only GBM IDH-mutant and GBM with oligodendroglial components seem to have a better prognosis in respect of the others [3,12,13]. Nowadays with the innovations introduced by molecular biology, the effect on overall survival (OS) of many molecular parameters has been investigated, such as p53, epidermal growth factor receptor (EGFR), IDH and MGMT. Some of these parameters demonstrated an interaction with the adjuvant treatment, thus significantly influencing survival [14]. The role surgery, or more precisely of the extent of resection (EOR), to date, remains unquestioned [15], although surgery can obtain totally different results in regards to the EOR, depending on the plethora of different techniques developed and introduced to maximize the resection while sparing the function, such as awake surgery, intraoperative neuromonitoring (IoN) and intraoperative neuropsychological testing (IoNT), intraoperative imaging methods and even hypnosis aided awake surgery (HAS). The resulting panorama is an incredible number of possible variables that can play a role in determining the incidence of LTS in a cohort. In this study we want to show a wide amount of clinical, radiological and molecular variables influencing the OS.

### 1.2. Purpose of the Present Investigation

The aim of the present investigation is therefore to analyze our retrospectively acquired database of LTS GBM patients treated in our institution in the period ranging between 2014 to 2016 and to compare their outcomes to those of our entire surgical GBM population, in order to understand and report the specific weight of a wide amount of variables in influencing the OS. A special focus was paid to the possible existence of correlations between the aforementioned GBM subtypes and the oncologic outcomes of the LTS patients, using the new classification together with other possible associated factors like entity of tumor removal and adjuvant treatments. 

## 2. Patients and Methods

### 2.1. Participants and Eligibility

We performed an institutional retrospective review of a consecutive series of surgically-treated patients suffering from histologically confirmed GBM, operated on in our department. We collected a total of 177 patients. Histological diagnoses were performed according to the updated version of the WHO guidelines [12]. We selected a total of 177 patients affected by newly diagnosed GBM who underwent surgery, radiation and chemotherapy in our institution in the period ranging between January 2014 and December 2016 meeting the following inclusion criteria: Patients were included in the study if their pre- and post-operative MR imaging was either performed at our institution or available on the picture archiving and communication system (PACS) for review.Patients were included if, in the postoperative period, they could undergo a standard Stupp protocol starting from the 30th–35th day after surgery as follows: Radiotherapy (60 Gy delivered in 30 fractions of 2 Gy/day, 5 days a week for 6 weeks) and concomitant oral chemotherapy with temozolomide (75 mg/m^2^ of body surface 7 days a week, from first to last day of radiotherapy, no more than 49 days). After a break of 20–25 days, about 12 cycles of Temozolomide (200 mg/m^2^ for 5 days every 28 days) were administered [9].Patients were included if they received standard conformational planning with a linear accelerator (LINAC), no stereotactic radiosurgical treatment was performed.Once the progression of the disease was noticed, the patient and the relevant imaging were referred again to our attention, to evaluate the feasibility of a second surgery or to address the patient to a second line of adjuvant treatment.The estimated target of the surgical procedure was the total or subtotal resection of the lesions: no biopsies were included.All the patients included in the study were newly diagnosed GBM at their first surgery; operating on recurrences makes a complete difference.Incomplete or wrong data on clinical, radiological and surgical records and/or lost to follow-up.

All the patients who met the aforementioned inclusion and exclusion criteria, were assigned on the ground of the survival parameters to the following two subgroups: Patients classified as LTS: experiencing an OS of at least 24 months or longer.Patients classified as short term survivors (STS): experiencing an OS of less than 24 months.

For all the included patients we recorded age, sex, location, tumor volume, clinical onset, IDH, Ki67, p53 and EGFR expression status. In particular, the specimens used in this study were examined for IDH mutation. Immunohistochemistry with Ki67, EGFR, ATRX and antibody anti-IDH1 R132H (Dianova, DIA H09; 1:50) was routinely performed in the Department of Neuropathology of our University Hospital. Overall survival was recorded in months; it was measured from the date of diagnosis to date of death or date of last contact if alive. Clinical information was obtained via the digital database of our institution, whereas OS data were obtained by telephone interview. A special focus was on the Karnofsky performance status (KPS) results: such parameter was considered, as previously observed [16], as associated with survival. In particular, it was recorded in three different moments: (1) Before surgery, (2) at 30 days after surgery and (3) at the end of the adjuvant treatment (the moment of the last outpatient evaluation). 

All the patients included underwent a preoperative brain magnetic resonance imaging (MRI) scan included a high-field 3 Tesla volumetric study with the following sequences: T2w, FLAIR, isotropic volumetric T1-weighted magnetization-prepared rapid acquisition gradient echo (MPRAGE) before and after intravenous administration of paramagnetic contrast agent; diffusion tensor sequences (DTI) with 3D tractography and functional MRI (fMRI) completed our protocol for what concerns gliomas affecting eloquent locations. The volume of the contrast-enhancing lesion was calculated, drawing a region of interest (ROI) in a volumetric enhancing post-contrast study weighted in T1 (a multi-voxel study), conforming to the margins of the contrast-enhancing lesion with software Osirix [17].

All the procedures were performed with an infrared-based Neuronavigator (Brainlab, Kick^®^ Purely Navigation), in a standard neurosurgical theatre, with a standard operative microscope (Leica, model OH4). In the first postoperative day, as routine, the patients underwent a volumetric Brain MRI scan to evaluate the EOR. 

For both subgroups (patients suffering from Type I and Type II tumors), in the case of a non-eloquently located lesion, a standard total intravenous anesthesia protocol with Propofol and Remifentanil has been used. For lesions involving the motor and language-related functional cortices, a standard full awake surgery protocol was routinely performed with the aid of intraoperative neuromonitoring realized with use of bi- and mono-polar stimulating probes, respectively, for the cortical and subcortical mapping. If intraoperative neuromonitoring or awake surgery were performed, no muscle relaxants were administered.

In general, it was intraoperatively judged necessary to stop tumor excision when:The white matter appeared free of disease in any aspect of the surgical cavity.Despite a directly visualized or navigation proven remnant, neuromonitoring or intraoperative neuropsychological testing outlined a risk for postoperative motor morbidity.

All procedures performed in studies involving human participants were in accordance with the ethical standards of the institutional and/or national research committee and with the 1964 Helsinki Declaration and its later amendments or comparable ethical standards. This article does not contain any studies with animals performed by any of the authors. Informed consent was obtained from all individual participants included in the study. The patient has consented to the submission of this review article to the journal.

## 3. Data Sources and Quantitative Variables

The extent of resection (EOR) was determined through a comparison between the MR images obtained before surgery and the first early MRI after surgery. EOR was calculated as a percentage by comparing the preoperative and early postoperative imaging, with the aforementioned software. Gross total resection (GTR), was defined as a confirmed reduction of the preoperative volume of the tumor of at least 95%, conversely, a near or subtotal resection was the surgical result on radicality (NTR/STR).

In the case of GTR, “tumor progression” was defined as the first MRI scan demonstrating the presence of pathologically enhancing tissue, characterized by an MRI pattern (relying mostly on perfusion-weighted imaging) inconsistent with a cerebral radiation injury (which is, in fact, a “pseudoprogression”). In case of incomplete resections (<95% volume reduction) a volumetric increase of the residual disease detected at the first postoperative MRI scan was considered disease progression. 

A close-range dedicated neuro-imaging follow-up program was routinely performed in our institution. This program included:A standard early (maximum 24 h after surgery) postoperative volumetric brain MRI.At approximately one month from surgery (25–35 days) a volumetric brain MRI scan was repeated for a first-step follow-up control and to provide information for the radiation treatment planning.After the end of irradiation, a volumetric brain MRI scan was performed every three months.

At every radiological reevaluation, we performed a complete outpatient clinical and neurological reevaluation. 

Generally, the treatment was considered to be stopped when disease showed volumetric progression despite the second line of adjuvant treatment. Both subgroups received surgical and adjuvant treatment with the same operative microscope, similar infrared-based neuronavigation system, similar microsurgical instruments, the same microsurgical technique, the same adjuvant treatment and follow-up program.

### 3.1. Statistical Methods

The sample was analyzed with SPSS version 18. A comparison between nominal variables was made with the Chi^2^ test. EOR, OS and progression-free survival (PFS) means were compared with one-way and multivariate ANOVA analysis along with contrast analysis and post-hoc tests. Kaplan–Meier survival analysis assessed survival. Continuous variables correlations have been investigated with Pearson’s Bivariate correlation. The threshold of statistical significance was considered *p* < 0.05.

### 3.2. Potential Source of Bias and Study Size

We addressed no missing data since incomplete records were an exclusion criterion. A potential source of bias is expected from exiguity of the sample, which nevertheless, in regard to the endpoints selected, presents an excellent post-hoc statistical estimated power (1 − β = 0.90 for α 0.05 and effect size 0.59), thus providing extremely reliable conclusions.

The informed consent was approved by the Institutional Review Board of our institution. Before surgical procedure, all the patients gave informed explicit written consent after appropriate information. Data reported in the study have been completely anonymized. No treatment randomization has been performed. This study is perfectly consistent with the Helsinki Declaration of Human Rights.

## 4. Results

In the first group, we retrospectively reviewed the clinical, radiological and surgical records of 177 patients operated on for craniotomy and resection of GBM in the period ranging between 2014 and 2016. The total amount of patients belonging to the LTS subgroup was 30 (16.94%): 20 males and 10 females (M:F ratio: 2:1), the average age was 59.4 years ± 7.69 (range 29–81 years), the median was 61 years. The presenting symptoms were: seizures in eight patients (26.6%), motor deficits in five patients (16.6%), sensory deficits in six patients (4.9%), visual disturbances in one patient (2.4%), cephalalgia in eight patients (26.6%) and incidental diagnosis in two patients (6.7%) (Table 1). The average KPS was 89 ± 13.70 (range 70–100), the mean interval between onset of symptoms and diagnosis was 3 months (range 1 week to 6 months).

Two patients (6.7%) underwent intraoperative brain mapping procedures in awake surgery for lesions involving the motor, primary sensory or language areas. In this group an IoN realized by means of MEP and SSEP was employed. Macroscopic GTR was achieved for 28 patients (93.3%); NTR for 1 patient (33%); and STR removal in 1 patient (3.3%).

The average follow-up was 3 years: nine patients (30%) are currently still alive. The median survival of the entire LTS subgroups was 32.40 months. Among the nine currently alive patients, four (44.44%) do not demonstrate relapses, whereas five (55.55%) experienced a recurrence of the disease: One of the latter underwent four relapses in the 5 years following the initial surgical procedure and is currently managed with an individual chemotherapy regimen. Among the 21 patients who died, 19 died of recurrent disease no longer treatable with surgery and no longer responding to alternative chemotherapy regimens or any form of radiation therapy.

### 4.1. Patient-Related Factors

Among the patient-related factors, we considered the effect of the following variables: age, sex, KPS trend in the pre- and post-operative period and the KPS score at the last evaluation and the clinical onset.

Younger age demonstrated an overall statistically insignificant trend to association with prolonged survival (61.16 versus 59.4 years, *p* = 0.708). Nevertheless the subgroup of patients who experienced an OS longer than 30 months presented an average age of 57.35 which, compared to the 61.16 of the STS group, demonstrated an increase to the statistical significance (*p* = 0.202, Figure 1). Sex did not show a statistically significant association either (*p* = 0.128).

Functional status proved to be a strong predictor of LTS evolution: In particular, the preoperative KPS score is associated with LTS patients, along with the postoperative KPS score. KPS at last evaluation presents no statistically significant difference between STS and LTS subgroups (KPS preoperative, postoperative and at last evaluation significances are *p* = 0.010, 0.163 and 0.721 respectively, Figure 2). In regard to the clinical onset, Chi^2^ analyses ruled out direct correlation between headache, seizures, language, motor, sensory, visual and gait disturbances (*p* = 0.159–0.544). Nevertheless, when specifically analyzed by means of a univariate ANOVA analysis, it was possible to retrieve a statistical association between a reduced PFS and language deficit, and reduced OS and motor deficit (*p* = 0.061 and *p* = 0.032 respectively, Figure 3A,B).

### 4.2. Tumor-Related Factors

From a molecular perspective, IDH mutation was strongly associated with a significant survival advantage, consistently with the ongoing evidence found in the literature (*p* = 0.008); Ki67 expression was associated with a shorter OS and PFS (*p* = 0.051 and 0.008, respectively). EGFR and p53 mutations did not show a significant association with the survival parameters, although presented an interesting reciprocal association between their incidence in our cohort (r = 0.334; *p* = 0.001). A multivariate and univariate ANOVA analysis confirmed that this interaction promotes a statistically non-significant survival advantage (Figure 4). To outline the specific weight of the different variables aforementioned on OS, we completed the analyses with a stepwise decomposition of the variance obtaining a factorial scaling as follows:p53 mutated EGFR mutated, Ki67 >20%.p53 mutated and EGFR mutated, Ki67 <20%.p53 mutated, EGFR wild-type, Ki67 <20%.p53 wild-type, EGFR mutated, Ki67 <20%.p53 wild-type, EGFR mutated, Ki67 >20%.p53 mutated, EGFR wild-type, Ki67 >20%.p53 wild-type, EGFR wild-type, Ki67 >20%.p53 wild-type, EGFR wild-type, Ki67 <20%.

We obtained a statistical inference through which was possible to outline the definitive weight of Ki67% in determining OS and PFS (*p* = 0.042 and *p* = 0.028). Moreover, it was possible to outline a survival trend, in a “scale” fashion according to the aforementioned molecular patterns (Figure 5). Classes 1 and 2 present a strongly significant survival advantage in comparison to class 8, as retrieved in contrast and post-hoc analyses (*p* = 0.011). Furtherly we compared the isolated impact on survival parameters of the combined EGFR and p53 mutations with respect to the clear impact of a Ki67 < 20% value in determining the survival advantage (*p* = 0.001, Figure 6). We have also evaluated MGMT data for 53 patients (17 MGMT methylated for short-time survivors and 8 MGMT methylated for long-time survivors), but with no significative results (*p* = 0.397, Table 1).

### 4.3. Surgery-Related Factors

The role played by the EOR on survival parameters was extremely clear with a strong association between GTR and prolonged PFS (*p* = 0.001); conversely, the association between GTR and OS did not demonstrate a statistical significance (*p* = 0.115, Figure 7). The side of the lesion did not demonstrate a statistically significant association with OS or PFS, nevertheless, generally midline and one multifocal lesion proved to be associated with a worse oncologic outcome, both in concerns to OS (both *p* = 0.001, Figure 8). Although negatively associated with PFS (r = −0.345, *p* = 0.001), it was not possible to identify a significant statistical difference between the incidence of LTS evolution and preoperative volume of the lesion; in any case, this parameter was associated both to PFS and EOR (both *p* = 0.001). The only location associated with a survival reduction was the involvement of corpus callosum, which, in respect to the remaining locations, was significantly associated with a shorter OS (*p* = 0.061, Figure 9 and Figure 10).

## 5. Discussion

The long-term survival of patients suffering from GBM is not a common finding. Improved survival of 24 months was found for the 16.9% of patients operated on in our institution. This percentage, although higher than reported by most of the authors, however, represents a small proportion of patients affected by this condition [4,6,18]. The poor prognosis is related to several factors, such as the aggressive nature of this disease, which often jeopardizes the feasibility of a real radical surgery, the presence of the blood–brain barrier (BBB) as a natural obstacle to the intracranial penetration of most of the conventional chemotherapeutic agents and the intrinsic refractoriness of GBM with respect to most cytotoxic agents [19,20,21]; Young age, female sex, a high KPS (>70) at diagnosis and prolonged PFS are the main ascertained patient-related factors. As most of the clinical series shows, age seems to play a critical role: LTS-patients have an average age of 45 years while non-LTS-patients of 60. [2,3,4,5,18,22]. Our study confirms some differences, with an average age of 57.35 for the LTS group which compared to the 61.16 of the STS group demonstrated to increase the statistical significance (*p* = 0.202, Figure 2). Sex did not show a statistically significant association either (*p* = 0.128) and so it is not confirmed, in our experience, a supposed protective effect of female hormones and/or the presence of a tumor suppressor gene located on X chromosomes reported in literature [22,23,24]. Finally, the association between LTS, high KPS at diagnosis and prolonged PFS is intuitive and stresses the importance of both the clinical and functional status of the cancer patients: In our experience, the average KPS was 80 and 78.6% of the patients currently alive have not shown signs of recurrence of the disease at a follow-up period.

Nevertheless, the LTS evolution is mainly observed in cases of GTR, thus confirming the role of surgery as a milestone in the management of GBM. Stummer et al. reported a median survival of 11.8 months in patients with a postoperative residual tumor, as opposed to 16.9 months in patients in which the remnant is not detectable. [25]. In general, the goal of the surgical treatment should be to maximize the EOR, while sparing the function, whereas adjuvant radiotherapy is aimed at reaching a regional control of disease prolonging the PFS [26].

In May 2009, Stupp et al. reported the final results of their study, they confirmed the effectiveness of the Temozolomide–radiotherapy treatment and shed a definite light over the critical role played by the level of methylation of the MGMT gene promoter [27]: Temozolomide plus radiotherapy improves survival in patients presenting lesions both with or without the methylated MGMT promoter, but the benefit on PFS is significant only for patients with promoter methylation [27]. Radiation therapy “alone” seems to increase the cancer cells′ radioresistance through *Akt* gene activation, mediated by the concurrent activation of EGFR [28,29,30].

We performed a thorough literature review concerning all the possible positive prognostic factors not included in our study, finding as possibly critical the role played by cancer stem cells (CSCs, CD33+). These cells constitute a small fraction of the tumor population and present “migration” tendencies (which helps to explain the infiltrative nature of GBM); CSCs are able to give rise to CD33- differentiated clones, demonstrating a higher potential growth rate thus contributing to the genesis of the “tumor-bulk” [31,32,33].

EGFR and p53 mutations did not show a significant association with the survival parameters in our cohort, in fact it was the opposite, they even seem to reverse the normal prognostic meaning that we usually associate, our series is complete, but not so large to consider a single variation of these parameter related to OS, we suppose that some different types of analysis and molecular factors have to be considered in the future and in the next classifications, EGFR and p53 are actually useful, but not sufficient, to understand the behavior of a GBM at the time of diagnosis. A wide amount of studies focused on the analysis of tumor cells genetic expression: More or less extensive deletions of DNA stretches involving different chromosomes (1p, 6q, 9p, 10p, 10q, 13q, 14q, 15q, 17p, 18q, 19q, 22q, Y) [34,35,36] have been investigated, but their role remains controversial. Combined 1p–19q loss of heterozygosity seems be a positive prognostic factor [4,5], because of its association with the presence of an oligodendroglial component in the contest of a GBM [11].

Many other genes seem able to modulate the biological behavior of glioblastoma cells. The *PTEN* gene should play an important role and the non-mutated form expression seems to be associated with a better response to treatment. It does not occur in the case of Akt expression [2,34,37,38,39]. This gene down-regulates PI3K, dephosphorylating PIP3: In case of its mutation, the high levels of PIP3 activate PI3K, that hyperphosphorylates Akt, with important effects on the cell proliferation and invasion. The *EGFR* gene amplification was claimed in 40–60% of primary glioblastomas, rarely in the secondary ones [34]. the most common mutation gives rise to the EGFRvIII variant, present in 20–50% of the cases of amplification. Both the amplification and mutation appear to be negative prognostic factors [4,10,37].

A limited number of investigations paid attention to the possible prognostic value of cytological and histological features. Nafe et al. reported that the main cytological difference between STS and LTS is the increase of the nuclear density in necrotic areas of the lesions, resulting in a reduction in the internuclear distance in STS patients [40]. Finally the role of Ki67, the number of mitosis and extent of necrosis as a negative prognostic value is uncertain [18,40]. Positive prognostic factors seem to be secondary and giant cells GBM subtypes [4,5,22] and the presence of oligodendroglial differentiation areas [11,35,41].

In recent years, numerous studies attempted to identify the factors underlying reasons for the prolongation of OS in LTS patients. Each time the genetic profiles and histological characteristics of the lesions, the clinical features of patients and the impact of different clinical treatments were analyzed. Although the impact of several treatment-related prognostic factors is well recognized, the precise identification of the subgroups of patients with an expected survival greater than 2 years remains, to date, controversial.

## 6. Conclusions

Despite the clinical work and research, the prognosis of GBM patients remains dismal. The LTS evolution is uncommon, and results from the combination of a wide number of concurrent therapy-, tumor- and patient-related factors. In our experience, LTS is associated with a GTR of tumor correlated with EGFR and p53 mutations regardless of localization, and poorly correlated to dimension.

We suppose that a standard molecular analysis performed (IDH, EGFR, p53 and Ki-67) are not sufficient to predict the behavior of a GBM regard to OS and a deeper understanding of the meaning of the different genetic alterations in the DNA of cancer cells, and a study with MGMT data included, will allow a more accurate prognostic stratification of the single patient resulting in a patient-specific therapeutic approach.

## Figures and Tables

**Figure 1 diagnostics-09-00209-f001:**
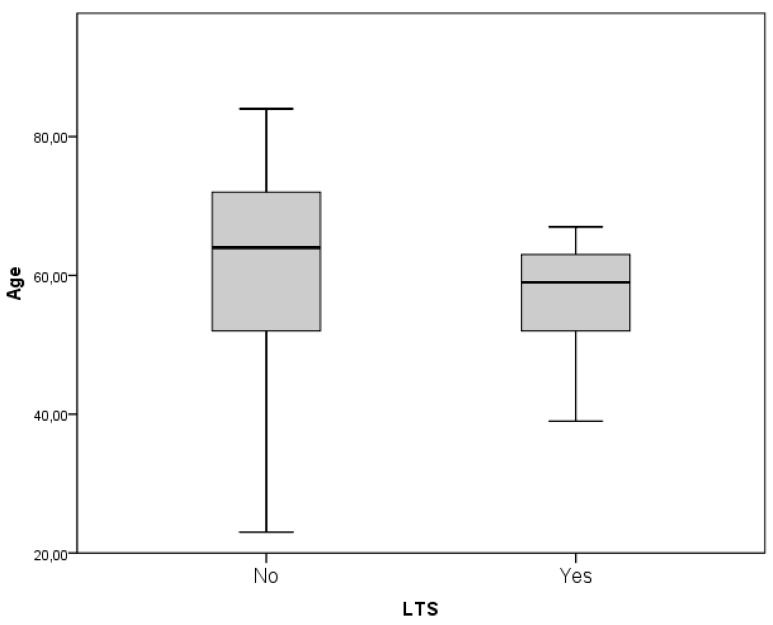
the subgroup of patients who experienced an overall survival (OS) longer than 30 months presented an average age of 57.35 which compared to the 61.16 of the short-term survival (STS) group demonstrated to increase the statistical significance (*p* = 0.202).

**Figure 2 diagnostics-09-00209-f002:**
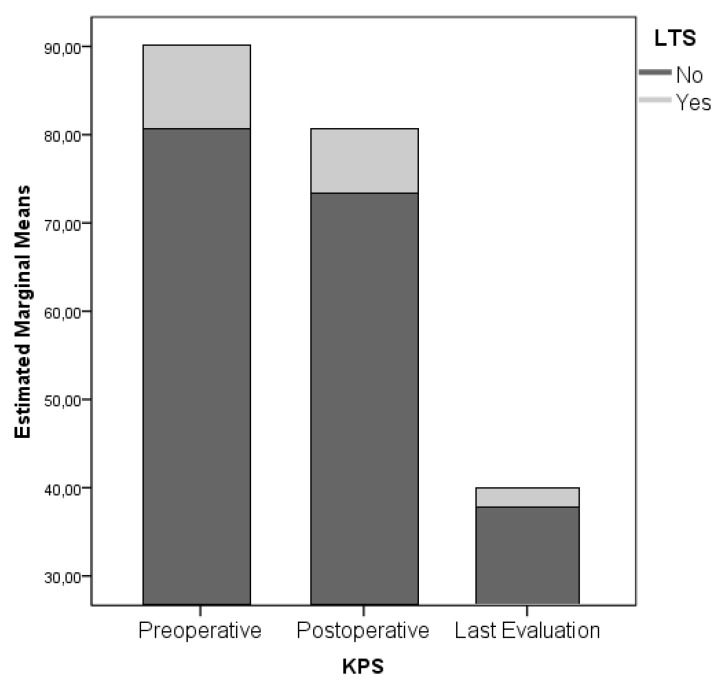
The preoperative Karnofsky performance status (KPS) score is associated with long term survival (LTS) patients, along with the postoperative KPS score. KPS at last evaluation presents no statistically significant difference between STS and LTS subgroups.

**Figure 3 diagnostics-09-00209-f003:**
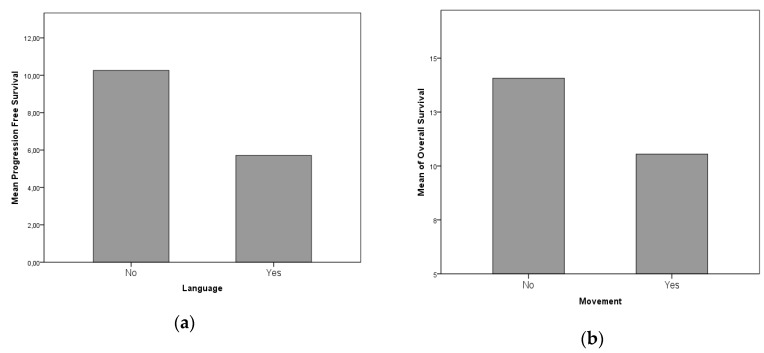
(**a**) and (**b**) with univariate ANOVA analysis, it was possible to retrieve a statistical association between a reduced progression-free survival (PFS) and language deficit, and reduced OS and motor deficit.

**Figure 4 diagnostics-09-00209-f004:**
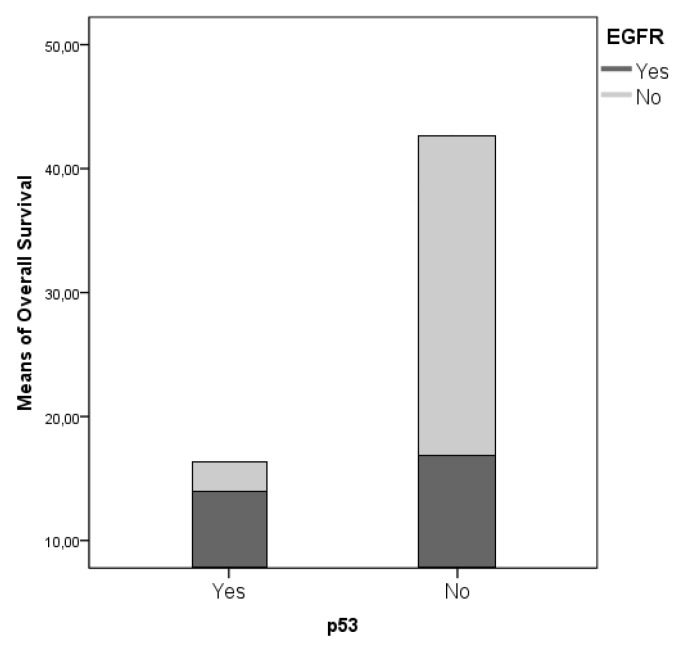
A multivariate and univariate ANOVA analysis confirmed that this interaction promotes a statistically non-significant survival advantage.

**Figure 5 diagnostics-09-00209-f005:**
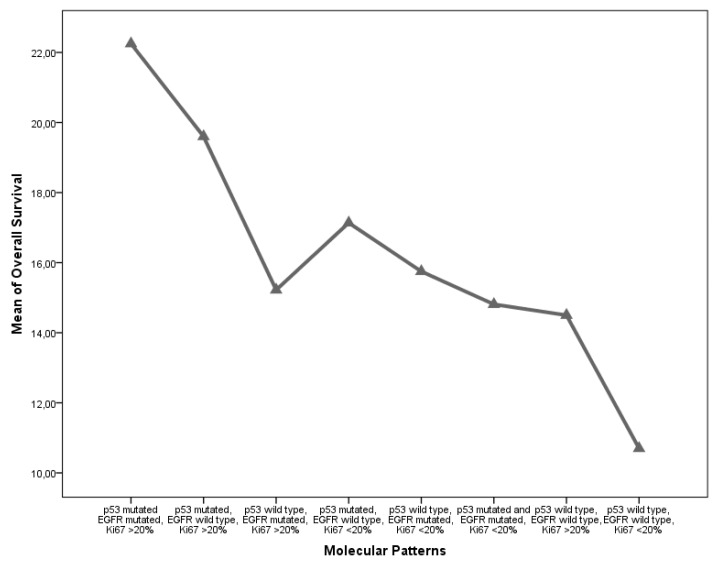
Classes 1 and 2 present a strongly significant survival advantage in comparison to class 8, as retrieved in contrast and post-hoc analyses.

**Figure 6 diagnostics-09-00209-f006:**
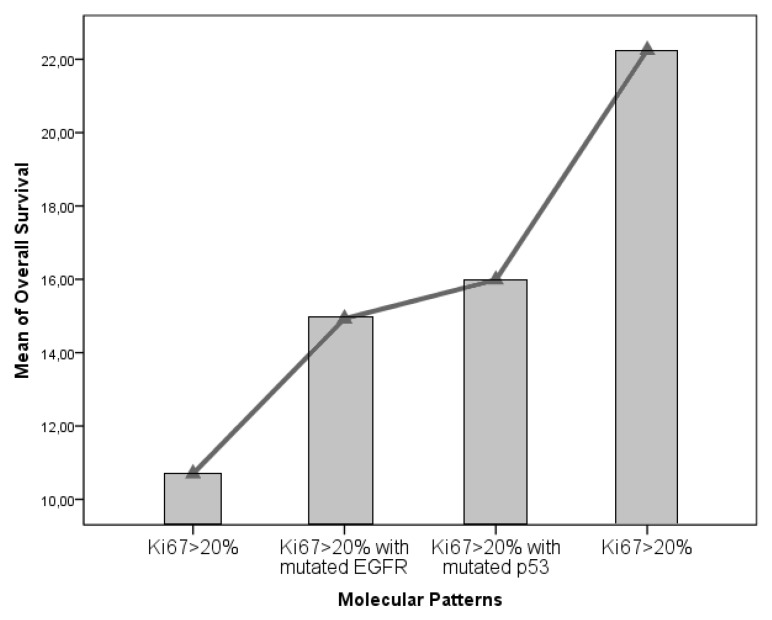
We compared the isolated impact on survival parameters of the combined epidermal growth factor receptor (EGFR) and p53 mutations with respect to the clear impact of a Ki67 < 20% value in determining the survival advantage (*p* = 0.001).

**Figure 7 diagnostics-09-00209-f007:**
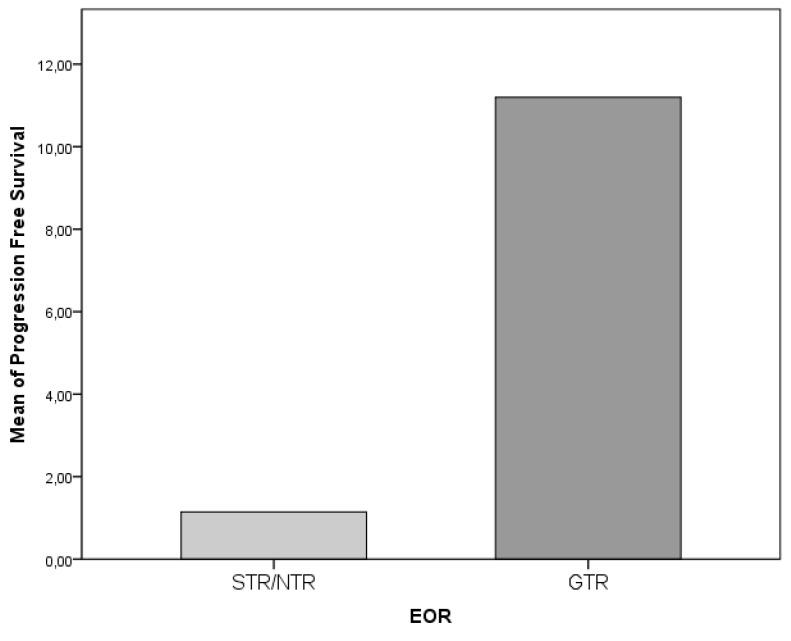
The association between gross total resection (GTR) and OS did not demonstrate a statistical significance (*p* = 0.115).

**Figure 8 diagnostics-09-00209-f008:**
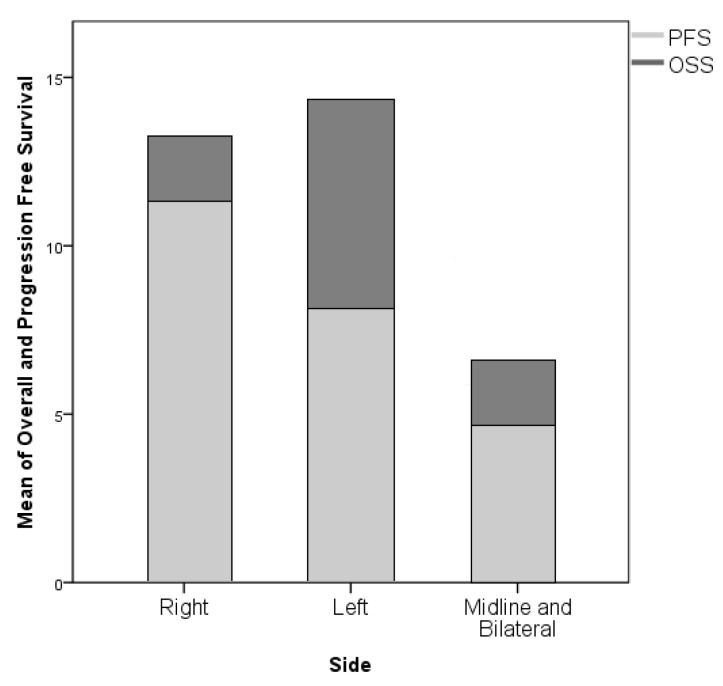
The side of the lesion did not demonstrate a statistically significant association with OS or PFS. Generally, midline and one multifocal lesion proved to be associated with a worse oncologic outcome.

**Figure 9 diagnostics-09-00209-f009:**
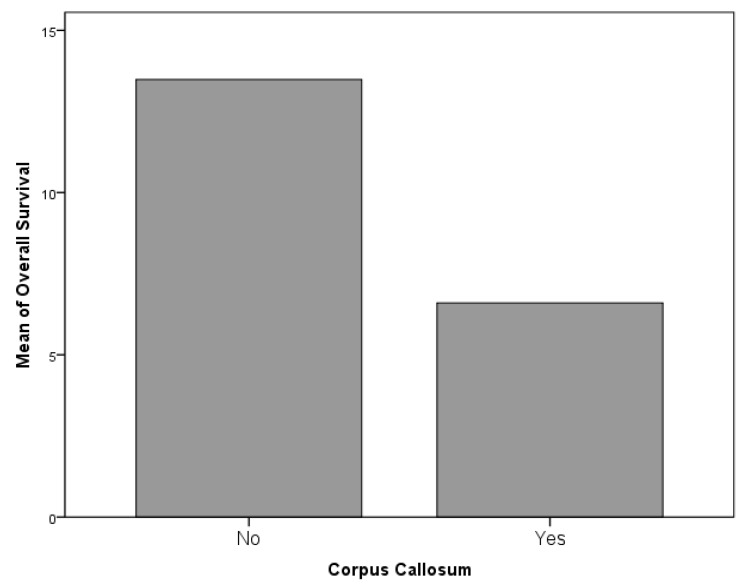
The only location associated with a survival reduction was the involvement of the corpus callosum, which, with respect to the remaining locations was significantly associated with a shorter OS (*p* = 0.061).

**Figure 10 diagnostics-09-00209-f010:**
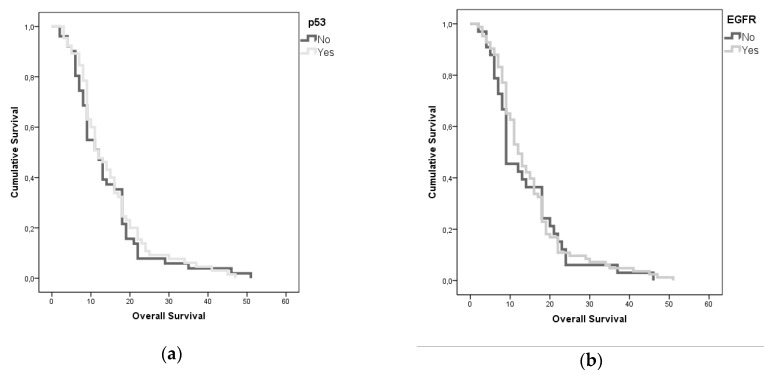
(**a**) and (**b**): A Kaplan-Meyer analysis shows the relationships between p53 and OS, and EGFR and OS.

**Table 1 diagnostics-09-00209-t001:** Patient’s demographics.

Contents	*n* = 177 Patients	Patients Collected between 2014–2016	*p* Value
Subgroup	LTS = 30	STS = 147	
Sex	Male *n* = 20–66.7% Female *n* = 10–33.3%	Male *n* = 78, 53.06% Female *n* = 69, 46.93%	0.128
Age	59.4 ± 7.69	61.16 ± 11.55	0.409
KPS at admission	89.0 ± 13.70	80.4 ± 12.41	0.010
Volume in cm^3^	24.2 ± 19.3	22.22 ± 18.4	0.676
Ki67 (%)	18.7 ± 10.9	26.4 ± 15.4	0.061
IDH Mutation status available in 166/177 pts	IDH Mutant 2/166 (6.7%)	IDH Mutant 0/166	0.027
EGFR overexpression status available in 149/177 pts	EGFR Overexpressed 9/26 (34.6%)	EGFR Overexpressed 35/124 (28.2%)	0. 333
MGMT Methylation status available in 53/177	MGMT Methylated 17 patients	MGMT Methylated 8 patients	0.397
p53Mutation status available in 150/177 pts	Mutant p53 Normal 18/27 (66.7%)	Mutant p53 66/124 (53.22%)	0.144
EOR	GTR 28/30patients (93.3%) STR 2/30 patients (6.7%)	GTR 133/147 patients (90.80%) STR 14/147 patients (9.20%)	0.468
KPS after Surgery	81.0 ± 25.11	73.9 ± 19.9	0.163
KPS at last Evaluation	39.5 ± 15.8	37.9 ± 17.6	0.721
Overall Survival	26.68 ± 7.1 months	10.8±4.8 months	0.001
Location	Frontal 14 (46.6%)	Frontal 47 (31.9%)	0.314
	Temporal 11 (36.6%)	Temporal 39 (26.5%)	
	Occipital 4 (13.3%)	Occipital 13 (8.8%)	
	Parietal 4 (13.3%)	Parietal 34 (23.1%)	
	Insular 2 (6.7%)	Insular 8 (5.44%)	
	Rolandic 2 (6.7%)	Rolandic 1 (0.7%)	
	Corpus Callosum 0 (0.0%)	Corpus Callosum 5 (3.4%)	
Side	Left 18 (60.0%)	Left 69 (46.9%)	0.743
	Right 12 (20.0%)	Right 66 (44.8%)	
	Midline 0 (0.0%)	Midline 8 (5.44%)	
	Multifocal 0 (0.0%)	Multifocal 1 (0.7%)	
Symptoms	Headache 8 (26.6%)	Headache 25 (17.1%)	0.115
	Seizures 8 (26.6%)	Seizures 41 (27.9%)	0.544
	Speech Disturbance 0 (0.0%)	Speech Disturbance 27 (18.4%)	0.531
	Motor Dysfunction 5 (16.6%)	Motor Dysfunction 35 (23.8%)	0.491
	Sensory Disturbance 6 (20.0%)	Sensory Disturbance 21(14.3%)	0.600
	Visual Deficit 1 (3.3%)	Visual Deficit 5 (3.4%)	0.423
	Incidental 2 (6.7%)	Incidental 4 (2.7%)	0.384

PFS: Progression Free Survival; OS: Overall Survival; SVZ: Subventricular Zone, KPS: Karnofsky performance status, EOR: Extent of Resection, GTR: Gross Total Resection, NTR/STR: Near Total/Subtotal Resection.

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
