# Peer review of "Long Term Survival in Patients Suffering from Glio-blastoma Multiforme: A Single-Center Observational Cohort Study"

_diagnostics, 2019, doi:10.3390/diagnostics9040209_

Round 1

Reviewer 1 Report

The manuscript is interesting but poorly written. There are many points that need to be clarified, especially in this case the results from the study seems conflict with the literature. I also don’t find the study to be novel or offer any new insight into GBM diagnosis. My detailed comments is as below:

- Classification of GBM has been updated extensively since 2007 to include both histology and molecular features. Furthermore, molecular subtyping has been proven to be significant. Please update the manuscript to reflect those recent advances.

- The patient demographics does not seem to mention EGFR mutations. However, the manuscript repeatedly refers to this as a positive prognosis indicator. Please clarify and elaborate on the possible mechanisms why EGFR mutation in this study seems to have the opposite effect in comparison to other published studies.

- Table 1 is very messy, inconsistent and sometime unintelligible (e.g. the p53 entry).

- Figure 3 seems like it has been scanned and cropped from a printed page. Please provide original, high quality graph.

- I makes no sense to use line graphs in Figure 4, 5, 6 and 8. Bar graphs are more appropriate.

- Please provide Kaplan-Meier plots showing survival of patients stratified by mutation EGFR/p53 mutation statuses.

- For all graph, please also plot all data points instead of just the mean. See: https://www.graphpad.com/support/faq/graph-tip-how-can-i-make-a-barcolumn-graph-that-also-shows-the-individual-data-points/

- Please proofread the manuscript to ensure consistent formatting (e.g. Ki67 vs ki67…). Certain phrases are capitalized needlessly (e.g. Cancer Stem Cells…). Also, it’s blood–brain barrier, and not usually referred as “Brain Blood Barrier”.

- Please provide a de-identified table of data used in this study.

Author Response

Classification of GBM has been updated extensively since 2007 to include both histology and molecular features. Furthermore, molecular subtyping has been proven to be significant. Please update the manuscript to reflect those recent advances.

We remarked, describes and citate the new classification to distinguish GBM IDH-wild type and IDH-positive;

The patient demographics does not seem to mention EGFR mutations. However, the manuscript repeatedly refers to this as a positive prognosis indicator. Please clarify and elaborate on the possible mechanisms why EGFR mutation in this study seems to have the opposite effect in comparison to other published studies.

EGFR and p53 mutations did not show a significant association with the survival parameters in our cohort, on opposite, they even seem to reverse the normal prognostic meaning the we associated, our series is complete, but not so large to consider a single variation of these parameter related to OS, we suppose that some different types of analysis and molecular factors have to be considered in future and in the next classifications, EGFR and p53 are actually useful but not sufficient to understand a behavior of a GBM at the time of diagnoses.

Table 1 is very messy, inconsistent and sometime unintelligible (e.g. the p53 entry).

We have re-defined the table.

Figure 3 seems like it has been scanned and cropped from a printed page. Please provide original, high quality graph.

We replaced the image with an high quality graph.

I makes no sense to use line graphs in Figure 4, 5, 6 and 8. Bar graphs are more appropriate.

We added a bar graphs.

Please provide Kaplan-Meier plots showing survival of patients stratified by mutation EGFR/p53 mutation statuses.

We added it

Please proofread the manuscript to ensure consistent formatting (e.g. Ki67 vs ki67…). Certain phrases are capitalized needlessly (e.g. Cancer Stem Cells…). Also, it’s blood–brain barrier, and not usually referred as “Brain Blood Barrier”.

General grammar correction were done

Please provide a de-identified table of data used in this study.

We added it

Reviewer 2 Report

the manuscript by Armocida et al entitled 'Long Term Survival in Patients suffering from Glioblastoma Multiforme: a Single-Centre Observational Cohort Study' is well written and and the analysis is convincing.

It would be better if author can include molecular and metabolic heterogeneity of GBM in their introduction section. It would further enhance manuscript quality by providing recent findings other than WHO classification.

Author Response

It would be better if author can include molecular and metabolic heterogeneity of GBM in their introduction section. It would further enhance manuscript quality by providing recent findings other than WHO classification.

We enlarge and describe molecular and new classification description in introduction;

Round 2

Reviewer 1 Report

The authors have addressed my concerns.